# Topological multiferroic order
# in twisted transition metal dichalcogenide bilayers

**Mikael Haavisto, Jose L. Lado and Adolfo O. Fumega**⋆

Department of Applied Physics, Aalto University, 02150 Espoo, Finland

⋆ adolfo.oterofumega@aalto.fi

## Abstract

Layered van der Waals materials have risen as powerful platforms to artificially engineer correlated states of matter. Here we show the emergence of a multiferroic order in a twisted dichalcogenide bilayer superlattice at quarter-filling. We show that the competition between Coulomb interactions leads to the simultaneous emergence of ferrimagnetic and ferroelectric orders. We derive the magnetoelectric coupling for this system, which leads to a direct strong coupling between the charge and spin orders. We show that, due to intrinsic spin-orbit coupling effects, the electronic structure shows a non-zero Chern number, thus displaying a topological multiferroic order. We show that this topological state gives rise to interface modes at the different magnetic and ferroelectric domains of the multiferroic. We demonstrate that these topological modes can be tuned with external electric fields as well as triggered by supermoiré effects generated by a substrate. Our results put forward twisted van der Waals materials as a potential platform to explore multiferroic symmetry breaking orders and, ultimately, controllable topological excitations in magnetoelectric domains.



# 1 Introduction

Twistronics has provided a new strategy to engineer correlated states stemming from the emergence of nearly flat moiré bands [1–6]. Twisted bilayer graphene represents an early example of this, displaying nearly flat bands close to 1° rotation that become strongly correlated [1–4]. Twisted graphene multilayers provide thus a correlated model with strong interactions that can be easily controlled via electronic gating, leading to both symmetry broken [7–11] and topological states [12–14]. Other layered van der Waals materials including transition metal dichalcogenides (TMDs) represent an excellent building block to engineer a twisted system [15–20, 20–23]. Specifically, TMDs monolayers can already display ordered phases [24,25] that can be combined to form a twisted system. Moreover, they provide a source for spin-orbit coupling interactions [26–28], which may drive a non-trivial topological character in the moiré system [16]. Experimental realizations of twisted transition metal dichalcogenide heterostructures have shown emergent magnetic and charge order [15–20, 20–23], including the emergence of strong magnetoelectric response [29] and a Haldane Chern insulator [30]. However, the potential emergence of multiferroic order in an artificial twisted system remains relatively unexplored.

Multiferroic materials are characterized by the simultaneous existence of more than one symmetry breaking [31–33]. These multiple symmetry breaking materials display a strong coupling between their different order parameters. For the particular case of electric and magnetic orders, a magnetoelectric coupling [34] provides a venue for the electric control of magnetic orders, a feature with a huge potential interest. Different multiferroic mechanisms have been studied over the past years, both in bulk compounds [35–37] and recently in two-dimensional monolayers [38–40]. In the realm of moiré materials, the strongly correlated states emerging in twisted systems provide an additional platform to artificially engineer multiferroics associated with the moiré length scale.

In this work, we show how a topological multiferroic order can be engineered in a twisted dichalcogenide bilayer. We start showing how twisted transition metal dichalcogenide homobilayers realize an effective correlated model in a staggered honeycomb superlattice. We show how the existence of competing long-range electronic interactions leads to the simultaneous emergence of ferrimagnetic and ferroelectric orders at quarter-filling. This multiferroic behavior is accompanied by a strong magnetoelectric coupling. Subsequently, we will analyze the necessary ingredients to turn this twisted multiferroic into a topological multiferroic. Finally, we show how the different ferroic domain walls that one can engineer in this topological system allow the magnetoelectric creation and control of topological Jackiw-Rebbi solitons. Our results put forward a strategy to obtain a multiferroic order in a twisted van der Waals heterostructure, and to exploit magnetoelectric control of multiferroic domains to engineer topological excitations.

# 2 Model

We start by describing the heterostructure, consisting of two layers of a transition metal dichalcogenide twisted forming a moiré pattern. The structure of the twisted TMD bilayer is shown in Fig. 1a, where each site corresponds to a transition metal atom. Two different inequivalent sites emerge: i) AA sites (yellow circles) where the transition metal atoms are perfectly aligned forming a triangular lattice and ii) two equivalent AB and BA sites (red circles) where the transition metal atoms are perfectly misaligned and form a staggered honeycomb



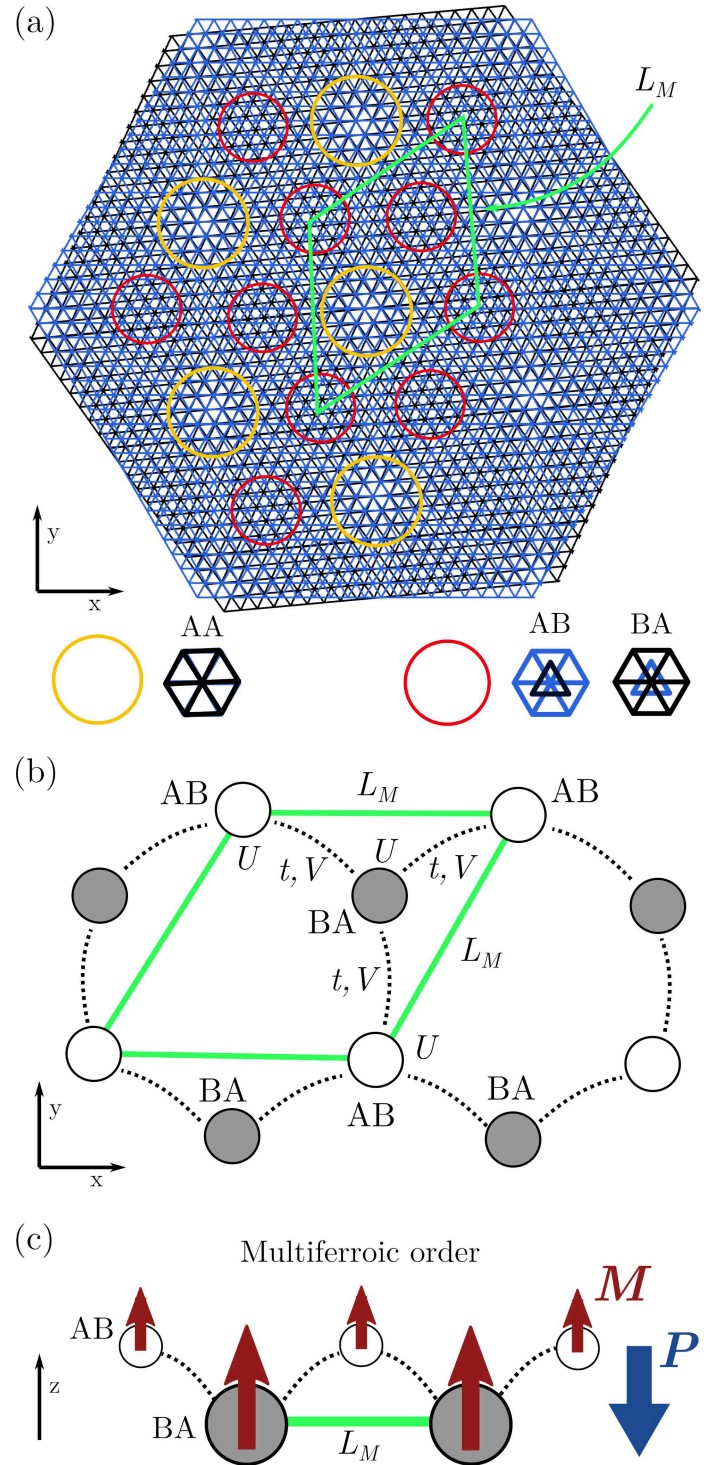

Figure 1: (a) Schematic of the twisted bilayer triangular lattice associated with the transition metal M of the $MX_2$. Sites AB and BA are structurally equivalent and display a staggered honeycomb lattice with a lattice parameter corresponding to the moiré length $L_M$. (b) Staggered honeycomb lattice model. Moiré unit cell in green and AB (BA) sites are depicted as white (gray) circles. First neighbor hopping $t$, on-site $U$, and first neighbor $V$ Coulomb interactions are schematically shown. (c) Emergent ferrimagnetic and fully gapped charge density wave orders from the quarter-filling staggered honeycomb model. In this state, the electric charge gets more localized in the BA sites which leads to an emergent electric polarization in the staggered honeycomb lattice.

lattice.[1] First principles and multiorbital Slater-Koster calculations [41–45] have shown the emergence of nearly flat bands in this twisted structure, stemming from the spatial modulation in the moiré unit cell. The emergent moiré mini bands feature localized states both in triangular and honeycomb lattices, corresponding to Wannier orbitals localized in the different stacking regions shown in Fig. 1a. In the following, we focus on the moiré mini-bands featuring a honeycomb lattice [41,46,47]. We focus on the regime in which the two sublattices of the effective moiré honeycomb model feature Wannier states localized in different layers, realizing an effectively staggered honeycomb model.[2] The Wannier Hamiltonian produced by the moiré pattern is written as

$$H = t \sum_{\langle ij \rangle s} c_{i,s}^\dagger c_{j,s} + U \sum_i c_{i\uparrow}^\dagger c_{i\uparrow} c_{i\downarrow}^\dagger c_{i\downarrow} + V \sum_{\langle ij \rangle ss'} c_{i,s}^\dagger c_{i,s} c_{j,s'}^\dagger c_{j,s'}, \tag{1}$$

where $t$ is the first neighbor hopping, $U$ the on-site Coulomb interaction and $V$ the first neighbor Coulomb interaction, $c_{i,s}^\dagger$ and $c_{j,s'}$ are the usual creation and annihilation fermionic operators for the Wannier moiré orbitals $i$ and $j$. As a reference, the effective value of $t$ for the twisted dichalcogenide system can be tuned from 1 to 50 meV depending on the twist angle [43,47–49]. It is worth noting that in the nearly flat regime, the effective model is dominated by first neighbor hopping due to localization of the Wannier states in specific points of the moire unit cell of transition metal dichalcogenides. The values of $U$ and $V$ range from 5 to 100 meV depending on twist angle and screening effects [49–52]. Our results focus on the regime in which $U, V$ are smaller than the separation between moire bands. Long-range interactions run between neighboring moiré Wannier orbitals $\langle ij \rangle$. Such electronic interactions stem originally from electronic repulsion of the d-orbitals of the transition metal dichalcogenide and generically long-range electrostatic repulsion. These interactions are directly projected in those nearly flat bands and are the ones effectively included in eq. (1). A schematic of the model is shown in Fig. 1b. We will focus on the quarter-filling limit, a regime that can be experimentally reached by electronic gating of the mini-bands.

The interacting model is solved using a self-consistent mean-field procedure including all the Wick contractions that allow magnetic symmetry breaking, hopping renormalization, and charge orders.[3] In the case of a multiferroic with electric and magnetic orders the order parameters are the electric polarization $P$ stemming from the staggered nature of the lattice:

$$P = d \left( \sum_s \langle c_{BA,s}^\dagger c_{BA,s} \rangle - \sum_s \langle c_{AB,s}^\dagger c_{AB,s} \rangle \right), \tag{2}$$

where $d$ is the vertical distance between sublattices that corresponds to the bilayer width, which we will take in natural units $d = 1$. The magnetization in the $z$-direction on each of the sites $M_\alpha$, ($\alpha = AB, BA$) is given by

$$M_\alpha = \sum_s \sigma_{ss}^z c_{\alpha,s}^\dagger c_{\alpha,s} = \langle c_{\alpha,\uparrow}^\dagger c_{\alpha,\uparrow} \rangle - \langle c_{\alpha,\downarrow}^\dagger c_{\alpha,\downarrow} \rangle, \tag{3}$$

and $M = \sum_\alpha M_\alpha$. A non-zero value on these order parameters induced by the interactions $U, V$ indicates the spontaneous emergence of the associated magnetic or charge order. A multifer-

---

[1]Note that including the effect of the chalcogen atoms in the twisted system will generate an inequivalence between AB and BA sites that will be effectively translated into a small sublattice imbalance in the staggered honeycomb model.

[2]The flat bands near the Fermi level or valence band edge will be the easiest to access experimentally via electronic gating, and hence this has been our subject of study.

[3]Anomalous terms related to the superconducting order are not included. The self-consistent calculations were carried out in the unit cell of the staggered honeycomb lattice in momentum space with a well converged 10×10 k-mesh. An initial guess with finite ferroic orders was used in the calculations.

roic behavior will occur when both order parameters $P$ and $M$ are simultaneously different than zero.

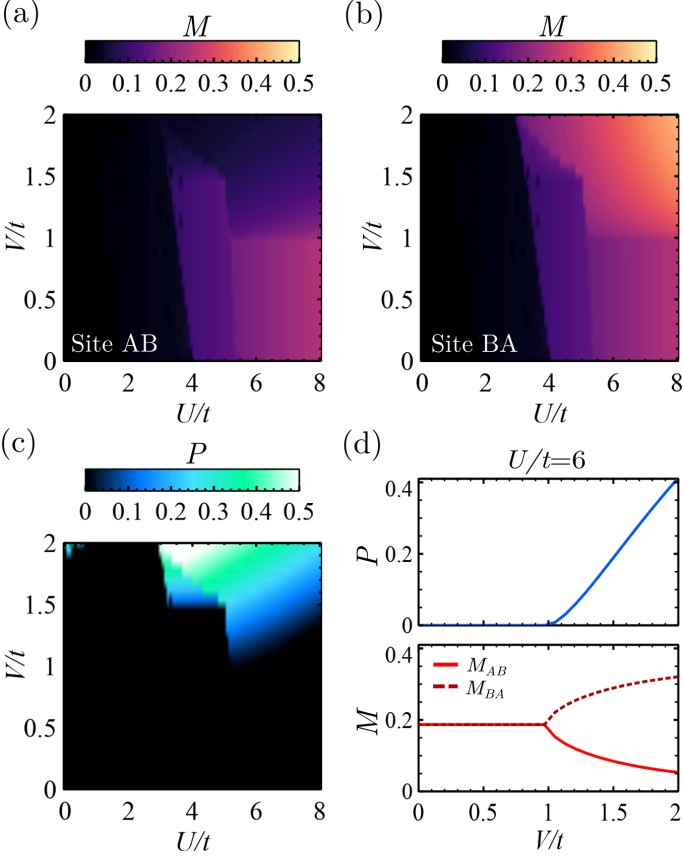

Figure 2: Emergence of the multiferroic order. (a,b,c) Phase diagrams as a function of the on-site $U$ and first neighbor $V$ Coulomb interactions for the different order parameters defined in the text: Magnetization $M$ on sites AB (a) and BA (b), and electric polarization $P$ (c). Magnetization is promoted by $U$ interactions, and $V$ interactions promote a sublattice imbalance that generates a finite electric polarization. A ferrimagnetic phase and a fully gapped charge density wave coexist in the top-right region of the phase diagram. This multiferroic behavior can be better seen in a cut at $U/t = 6$. (d) Order parameters as a function of $V/t$ for $U/t = 6$, $P$ ($M$) in the top (bottom) panel. For $V/t > 1$ both order parameters present a finite value and the magnetization on each site becomes inequivalent.

## 3 Multiferroic order from competing interactions

From a physical point of view, the quarter filling in the staggered honeycomb lattice offers a natural platform in which electronic interactions can lead simultaneously to the emergence of a charge order and a magnetization. Compared to the well studied half-filling case, in which electronic interactions lead to an antiferromagnetic insulator, the quarter-filling case allows the stabilization of charge order, thus leading to a net electric polarization in the staggered honeycomb lattice. The onsite Coulomb interaction $U$ leads to a magnetic Stoner instability, in the half-filling case this leads to an equal magnetization in absolute value in each of the sites, since there are 2 electrons for 2 sites. For the realistic case in which first neighbor Coulomb

interactions $V$ are smaller than onsite, $U > V$, a sublattice imbalance will not be promoted, since this is energetically unfavorable, i. e., an electron is already occupying each site. In the quarter-filling case only 1 electron is available for the 2 sites, the onsite interaction will promote a Stoner instability leading to magnetic order. However, in this case, for $U > V$, $V$ will be able to promote a sublattice imbalance, since the sites are not fully occupied by an electron. Therefore, the quarter filling case allows for the simultaneous emergence of magnetic and electric orders.[4]

The resulting phase diagram at quarter filling of the interacting Hamiltonian (eq. (1)) for the different order parameters is shown in Fig. 2. Figures 2ab show the phase diagram as a function of the on-site $U$ and first neighbor $V$ Coulomb interactions for the magnetization in each of the sites AB and BA respectively. It can be seen how on-site interactions promote magnetism in the system (right side of the phase diagrams). For $U/t > 4$ a spontaneous magnetization emerges in both sites of the lattice. Figure 2c shows the phase diagram for the electric polarization. It can be seen that the combination of on-site and first neighbor Coulomb interaction leads to the emergence of an electric polarization $P$ (top right corner of the phase diagram). The emergence of the electric polarization is a consequence of the spontaneous charge order, promoted by first neighbor Coulomb interactions, between sites AB and BA. In a staggered honeycomb lattice, a staggered charge order produces a net electric polarization in the perpendicular direction, leading to a ferroelectric dipole. In doped monolayer semiconductors spin-orbit coupling favors an out-of plane magnetic ordering, thus driving the magnetization in the z-direction [53].[5] Therefore, the simultaneous emergence of the magnetic and ferroelectric orders constitutes a multiferroic order in the system, like the one depicted in Fig. 1c.

The emergence of the multiferroicity can be rationalized in Fig. 2d. There, a plot of both orders $P$ and $M$ as a function of $V$ is shown fixing $U/t = 6$. It can be seen that for $V/t < 1$ a ferromagnetic order occurs where both sites display the same magnetization and, for the same $V$ values, no net polarization is present in the system. For $V/t > 1$ a stagger charge order is promoted as a function of $V$, creating a spontaneous electric polarization and leading the ferromagnetic order to a ferrimagnetic one. Associated with the charge order, the magnetization in each of the sites becomes different, a feature that is directly reflecting a magneto-electric coupling. The spin polarized situation and the stagger charge order can be observed in the band structure shown in Fig. 3a. At quarter-filling a gap opens due to Coulomb interactions, promoting a spin polarized situation and the charge gets more localized on the BA site, creating a sublattice imbalance. The onsite interaction $U$ plays also an important role in the stabilization of the CDW order. The relevance of $U$ stems from inducing a spin symmetry breaking, which in turn pins the chemical potential at the Dirac point at the majority spin channel. This pinning allows the first neighbor interaction $V$ to drive a CDW state, as such a symmetry breaking allows opening up a gap at the Dirac point. Therefore, onsite interactions cooperate with $V$ to drive CDW at quarter filling.[6] The multiferroic order that emerges from the interplay between $U$ and $V$ in the interacting Hamiltonian of eq. (1) is a combination of ferroelectric and ferrimagnetic orders.

The specific values for the electric polarization and magnetic moments will depend on the specific values of the electronic interactions $U$ and $V$ as shown in Fig. 2d. However, we can establish upper bounds for the electric and magnetic moments. For the magnetic moment, strong onsite interactions fix a value of 0.5 $\mu_B$ per moiré unit cell. For the electric dipole, the

---

[4]The three-quarter filling case would be similar to the quarter-filling one and it would also lead to a multiferroic order in the case of electron-hole symmetry.

[5]We also note that in the absence of such anisotropy, or in regimes in which it is not dominant, potentially non-collinear magnetic textures could emerge in the system, whose analysis goes beyond the scope of this manuscript.

[6]At low values of $U$ we have found in our analysis (but not shown, since $V > U$) that a CDW can also be formed for big values of $V$, i.e. $V > U$, but in this case the magnetic order does of course not emerge.

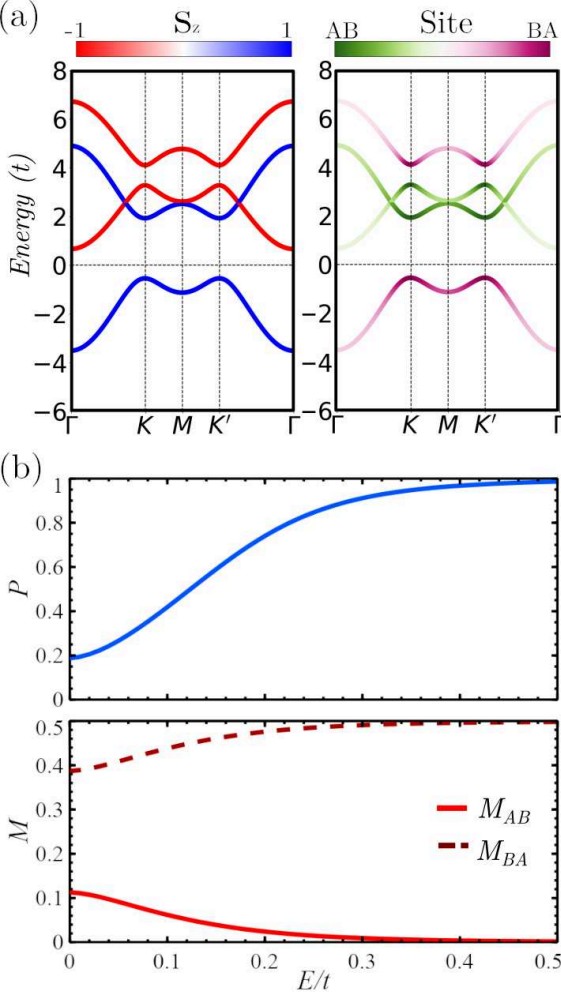

Figure 3: Calculations in the multiferroic regime at $U/t = 6$ and $V/t = 1.5$. (a) Band structure, the colormaps represent the eigenvalue of the operator: $s_z$ (sublattice site) in the left (right) panel. A spin polarized and sublattice situation can be identified. (b) Magnetoelectric coupling analysis. Evolution of the order parameters $P$ (top panel) and $M$ (bottom panel) as a function of an external electric field in the $z$-direction. Due to the strong magnetoelectric coupling, both the electric polarization and the ferrimagnetic order are controlled with the external electric field. The orders saturate at $E/t = 0.5$.

vertical localization of the Wannier orbitals plays also a fundamental role. The symmetry of the staggered honeycomb lattice guarantees that the emergence of a sublattice imbalance will produce a vertical electric polarization in the system, which relies on the vertical localization of the Wannier orbitals in each of the layers. In particular, in transition metal dichalcogenides, the Wannier states correspond to localized modes in each layer that appear due to local band bending due to the local stacking. This yields an upper bound for the vertical shift as the interlayer distance controlling the electric polarization of the system. Variations to this perfect localization will reduce the value of the ferroelectric polarization. Therefore, considering an interlayer distance of 4 Å and a perfect localization of the electron in one of the layers, we can establish an upper bound for the electric dipole on the order of ≈1 Debye per moiré unit cell.

A fundamental feature of multiferroic systems that display simultaneously electric and magnetic orders is the magnetoelectric coupling [34]. It refers to the existing coupling between the different order parameters associated with each of the ferroic states. In particular, a strong

magnetoelectric coupling allows controlling one of the orders by tuning the other one [36, 54]. Multiferroics whose microscopic mechanism leads to the simultaneous emergence of both orders are known as type-II multiferroics [32,33]. Their ferroic orders are not independent and therefore a strong magnetoelectric coupling is expected. In the system that we are studying multiferroicity arises due to the combination of competing electronic interactions, leading to the simultaneous emergence of a ferroelectric and a ferrimagnetic order. Therefore, we might expect a strong magnetoelectric coupling in this twisted multiferroic. In order to analyze that, we will include in the interacting Hamiltonian of eq. (1) the effect of an external electric field ($E$) perpendicular to the layers that directly couples to the ferroelectric dipole. This term takes the form

$$H_E = E \sum_s \left[ c_{AB,s}^\dagger c_{AB,s} - c_{BA,s}^\dagger c_{BA,s} \right], \tag{4}$$

which is nothing but a bias difference between Wannier orbitals in the AB and BA due to the staggered nature of the honeycomb lattice.

Figure 3b shows the evolution of the electric polarization $P$ and magnetization $M$ on each of the sites as a function of the electric field $E$. It can be seen that increasing the electric field increases the electric polarization as expected for a ferroelectric. Moreover, the module of the magnetization on each of the sites gets modified by the electric field, thus providing electric control of the ferrimagnetic order. These results show clear evidence of strong magnetoelectric coupling in the twisted multiferroic. It is worth noting that while the direction of the ferroelectric polarization is locked to the out-of-plane direction, the direction of the magnetization does not have any preferential direction. In the absence of spin-orbit coupling, only the modules of the ferroic order parameters are magnetoelectrically coupled in the interacting Hamiltonian that we are considering.

## 4 Topologically non-trivial moiré multiferroic

So far we have shown that a multiferroic behavior can emerge in the twisted system as a consequence of Coulomb interactions. Another important aspect associated to twisted TMDs is the possible realization of non-trivial topological states [55]. This section will aim to analyze the potential emergence of topological excitations associated with the twisted multiferroic system that we are studying. Transition metal dichalcogenides are well known to show strong spin-orbit coupling effects [56–58], which are known to account for the emergence of topological phase transitions [55]. In particular, the breaking of mirror symmetry in our system triggers the emergence of a Rashba SOC interaction in the low energy effective model [56,59]. In our twisted system, either the emergence of the out-of-plane electric polarization, or the inclusion of a substrate break mirror symmetry. Furthermore, the use of Janus transition metal dichalcogenides [60–62] would provide a built-in mirror symmetry breaking triggering a large intrinsic Rashba SOC effect [45]. The projection onto the low energy model of the effective Rashba SOC interaction takes the form:

$$H_R = i\lambda_R \sum_{\langle ij \rangle, ss'} \mathbf{z} \cdot \left( \sigma_{s,s'} \times \mathbf{d}_{ij} \right) c_{i,s}^\dagger c_{j,s'}, \tag{5}$$

where the sum runs over the first neighbors, $\lambda_R$ controls the strength of the Rashba interaction, $\mathbf{d}_{ij}$ represents a unit vector pointing from the site $j$ to $i$, $\sigma$ are the Pauli matrices and $\mathbf{z}$ is a unit vector along the $z$-direction.

We show the interacting electronic structure in Figure 4a, where the effect of the Rashba interaction in the multiferroic regime is observed. At low values of $\lambda_R$ (left panel), the insulat-

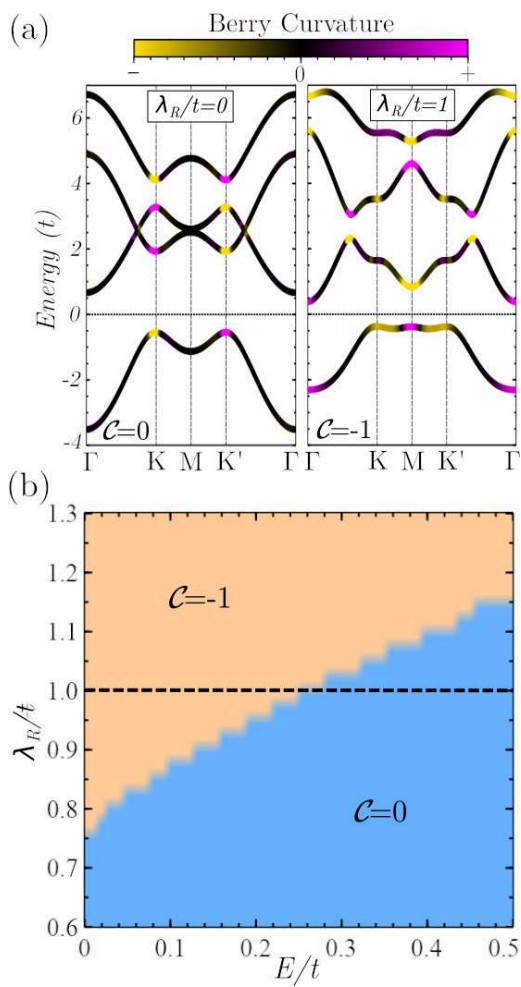

Figure 4: Topological analysis of the interacting Hamiltonian at $U/t = 6$ and $V/t = 1.5$. (a) Effect of the Rashba spin orbit coupling ($\lambda_R/t$) in the band structure, Berry curvature and Chern number $\mathcal{C}$: for $\lambda_R = 0$, $\mathcal{C} = 0$ (left panel) and for $\lambda_R = 1.0$, $\mathcal{C} = -1$ (right panel). (c) Phase diagram of the Chern number as a function of $\lambda_R/t$ and the external electric field bias $E/t$. At a given value of $\lambda_R$ (for instance $\lambda_R = 1.0$, dashed line) it is possible to modify the topological character of the system with an electric bias.

ing system has a Chern number $\mathcal{C} = 0$ and therefore displays a trivial topological character. At high enough values of $\lambda_R$ (right panel), a band inversion occurs, leading to a non-zero Chern number and turning the multiferroic system into a Chern insulator. This topological transition driven by Rashba spin-orbit coupling can be better analyzed in Fig. 4b. In that plot the interacting Hamiltonian (eq. (1)) is solved including both the electric field term (eq. (4)) and the Rashba SOC interaction (eq. (5)) in the multiferroic regime ($U/t = 6$ and $V/t = 1.5$). A phase diagram of the topological character (Chern number $\mathcal{C}$) as a function of the electric field $E$ and the Rashba SOC $\lambda_R$ is shown in Fig. 4b. The boundary between the orange ($\mathcal{C} = -1$) and blue ($\mathcal{C} = 0$) regions is where the band gap closes leading to a band inversion and consequently to a topological phase transition. Interestingly, in this phase diagram, we can see that the external electric field can cause a topological transition at a given value for the Rashba SOC (see the dashed line at $\lambda_R/t = 1$). This result shows that it is possible to control the topological character of this system via external electric fields. Therefore, this, together with the multiferroic character of the twisted system, will lead to a magnetoelectric control of diverse

topological excitations in the ferroic domain walls of this twisted system as we will address in the next section.

The value of the Rashba SOC will be determined by the transition metal and the ligand atoms. It is important to consider that the energy scales relevant in the twisted system are on the order of meV, both for hopping and Rashba SOC. Therefore, values on that order of magnitude for the Rashba spin-orbit coupling will be enough to obtain a non-trivial topological character as we can see from Fig. 4b.

# 5 Topological modes in multiferroic domains

In a conventional multiferroic material, different domains are expected to emerge when the sample is cooled at applied electric and magnetic fields. Furthermore, by local application of electric fields, junctions between different multiferroic domains can be engineered. In this section, we address the emergence of topological interface modes in the different domain walls that can occur in the topological multiferroic. In particular, we can encounter purely ferroelectric domains, purely ferrimagnetic domains, and simultaneously ferroelectric and ferrimagnetic domains. The topological character of each of these domains can be controlled by an external electric field as addressed in the previous section. This will allow the manipulation of the emergent interface-topological states. Furthermore, we will show how the underlying modulations induced by a substrate naturally lead to the emergence of topological modes associated to a new supermoiré length scale [63–65].

## 5.1 Tunable topological states in domain walls

Up to this point, we have based all the analyses on a full self-consistent solution of the interacting Hamiltonian of eq. (1). Since we will analyze now interfaces between different ferroic domains, in this section we will take a minimal effective mean-field Hamiltonian corresponding to the uniform limit, yet without solving self-consistently the interface problem. The effective mean-field Hamiltonian takes the form

$$H_N = \tilde{t} \sum_{\langle ij \rangle s} c^{\dagger}_{i,s} c_{j,s} + m \sum_{s} \left[ c^{\dagger}_{AB,s} c_{AB,s} - c^{\dagger}_{BA,s} c_{BA,s} \right] + \Delta_Z \sum_{i,s} \sigma^z_{ss} c^{\dagger}_{i,s} c_{i,s} + H_E + H_R , \quad (6)$$

where $\tilde{t}$ is the moiré hopping renormalized by the Coulomb interactions, $m$ accounts for the interaction-induced charge order accounting for the emergence of the electric polarization, and $\Delta_Z$ is the interaction induced exchange field associated with the spin polarization in the system. From the microscopic point of view, the three parameters depend on the interactions $U$ and $V$. At quarter-filling values with $m/t = 0.6$ and $\Delta_Z/t = 2.0$, the previous Hamiltonian is analogous to the mean-field result obtained self-consistently above. In the following we will include the external electric field $H_E$ (eq. (4)) and Rashba spin-orbit coupling $H_R$ (eq. (5)) in eq. (6). These terms account for the magnetoelectric coupling and the non-trivial topological character in the effective Hamiltonian.

A schematic of a device showing a minimal interface displaying topological excitations between ferroic domains is shown in Fig. 5a. In a sample with two domains, left (L) and right (R), external electric fields $E_L$ and $E_R$ allow controlling the topological character in each of them. Due to the existence of two valleys $K$ and $K'$ in the underlying electronic structure, a valley flux $\mathcal{C}_K$ and $\mathcal{C}_{K'}$ can be defined. Given that the Berry curvature is strongly localized around each valley, the total Chern number becomes $\mathcal{C} = \mathcal{C}_K + \mathcal{C}_{K'}$, and the so-called valley Chern number is given by $\mathcal{C}_V = \mathcal{C}_K - \mathcal{C}_{K'}$. In the absence of inter-valley scattering, the valley

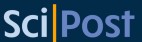

Figure 5: Magnetoelectric control of topological excitations in the domain walls of a topological multiferroic. (a) Schematic device that allows controlling the topological character of two connected ferroic domains, L (left) and R (right) domains, with external electric fields applied in the z direction, $E_L$ and $E_R$ respectively. The domains are semi-infinite in the y direction. The domain wall occurs at $y = 0$. In the x direction the lattice is periodic. Three different ferroic domain walls can occur: $FE$ purely ferroelectric, $FM$ purely ferrimagnetic and $FEM$ simultaneously ferroelectric and ferrimagnetic. The tables summarize the value of the valley Chern numbers $\mathcal{C}_K$ and $\mathcal{C}_{K'}$ as a function of the external electric fields for each kind of domain wall. The number of emergent interface states is also included. (b) Momentum resolved interface spectral function $A(\omega, k_x)$ for each of the domain walls $FE$ (top panels), $FEM$ (middle panels) and $FM$ (bottom panels) and for the different $E_L$ and $E_R$ values summarized in the tables of panel (a). The calculations were performed with the effective Hamiltonian (eq.(6)) considering $|m/t| = 0.6$, $|\Delta_Z/t| = 2.0$ and $\lambda_R/t = 0.8$.

is a good quantum number, each valley becomes independent and the valley Chern number $\mathcal{C}_V$ becomes quantized. Each valley can provide a topological flux of $\pm 1/2$, whose sign is determined by the combination of magnetic and electronic symmetry breaking, and acts as an independent topological source. Therefore, at the different ferroic domain walls (Fig. 5a) the emergence of an interface state at the $K$ ($K'$) point is determined by $\mathcal{C}_{K,R} - \mathcal{C}_{K,L}$ ($\mathcal{C}_{K',R} - \mathcal{C}_{K',L}$), i.e., the difference between the corresponding valley Chern numbers of domains R and L. The Chern number difference in each sector can be $\pm 1$ or 0. A non-zero value implies the emergence of a topological interface state in that valley sector. Therefore, since there are two independent sectors ($K$ and $K'$), the number of topological excitations that we can encounter at the interface can be 2, 1, or 0. Moreover, for finite values in the difference between valley Chern numbers ($\pm 1$), the sign at each valley determines the direction of propagation of the interface states. When the total number of interface states is 2, an opposite sign will indicate that both states counter-propagate, while the same sign will indicate co-propagation. Tables summarizing all the possible situations that can occur as a function of the external electric fields for each domain ($E_L$ and $E_R$) are shown in Fig. 5a. At huge value of the external electric fields ($E/t = \pm 1.0$ in the tables), valley Chern numbers are no longer good topological numbers due to intervalley mixing, and each domain is simply a trivial multiferroic. Consequently, no interface states will emerge in this limit situation. The Chern numbers are unaffected by the interface termination. For topological states with a valley Chern number, if the interface is too sharp a small gap could be opened driven by a strong intervalley scattering.

In order to demonstrate the emergence of topological interface states for each of the situations summarized in the tables of Fig. 5a, we have computed the momentum resolved interface spectral function $A(\omega, k_x)$. This is shown in Fig. 5b for all the different situations. In the case of purely ferroelectric domains ($FE$ top panels) increasing the value of the external electric fields drives the system from 0, 1, 2 counter-propagating and 0 interface states. In the case of purely ferrimagnetic domains ($FM$ bottom panels) increasing the value of the external electric fields drives the system from 2 co-propagating states to 1 state and ultimately 0 interface states at large bias. In the case of simultaneous ferrimagnetic and ferroelectric domains ($FEM$ middle panels) increasing the value of the external electric fields drives the system from 2 co-propagating states, 1 state, 2 counter-propagating, and ultimately at large bias 0 interface states. Therefore, these results prove the magnetoelectric control that can be achieved on the topological excitations of this topological multiferroic.

## 5.2 Topological domains in a supermoiré

So far, we have considered that the twisted bilayer dichalcogenide displays a single moiré pattern, whose length scale $L_M$ gives rise to the emergent staggered honeycomb lattice. However, in real moiré systems, a substrate is also present (as depicted in Fig. 6a). In particular, the lattice of an underlying substrate, such as boron nitride, gives rise to an additional supermoiré pattern between each dichalcogenide [63–65]. When projected on the nearly flat bands of the twisted system, this additional supermoiré pattern gives rise to a modulation of the original effective moiré superlattice. As sketched in Fig. 6b, from the point of view of the effective low energy model of the moiré system, this underlying supermoiré pattern gives rise to a modulation in space of the moiré model [66,67]. In the following, and for the sake of concreteness, the unit cell associated with the combination of the moiré pattern of the substrate and the twisted dichalcogenide will be denoted as the ultracell and will have an associated lattice parameter $L_{SM}$ that will be commensurate with the original moiré supercell with lattice parameter $L_M$ (see Fig. 6b). For computational reasons, the supermoiré potential is chosen to be commensurate with the supercell of the original moiré cell (staggered honeycomb unit cell) and it takes

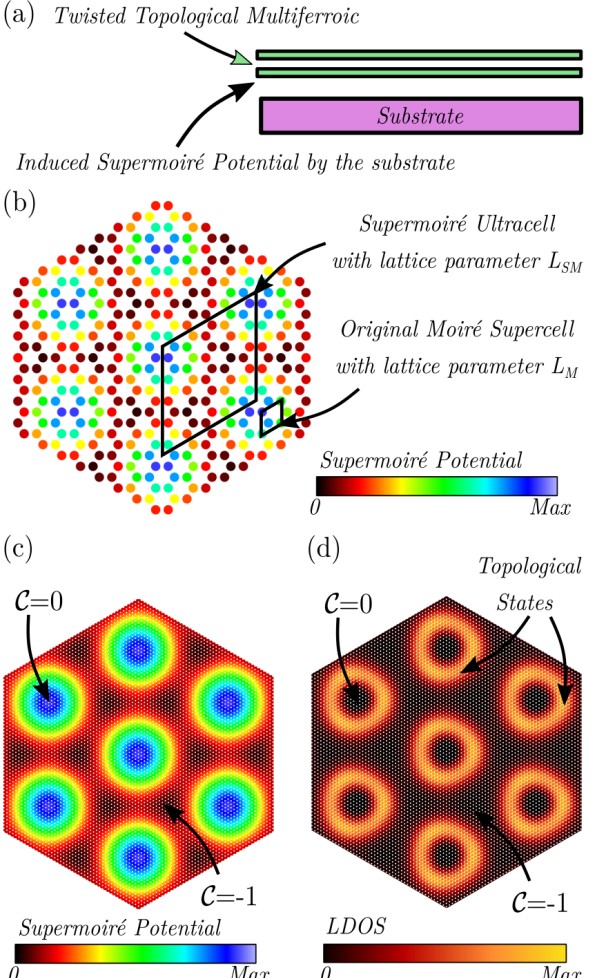

Figure 6: (a) Effect of a substrate on a twisted topological multiferroic. A super-moiré potential is induced in the twisted system by the substrate. (b) Sketch of the supermoiré potential on the staggered honeycomb lattice. The external field commensurates with a 5 × 5 ultracell of the staggered honeycomb lattice, $L_{SM} = 5L_M$. (c) Supermoiré potential of $L_{SM} = 30L_M$ used in the calculations. The electric potential creates a modulation with different topological-charge regions, $\mathcal{C} = 0$ on the maximum values of the potential and $\mathcal{C} = -1$ for the minimum values. (d) Local density of states (LDOS) at zero energy for the $L_{SM} = 30L_M$ supermoiré potential shown in panel (c). Circular topological states emerge at the boundaries of regions with different topological invariant, i.e, inside and outside of the emergent circles.

the following functional form:

$$E_{SM}(\mathbf{r}) = \sum_i \cos\left(\frac{\mathbf{b}_i \cdot \mathbf{r}}{n}\right), \tag{7}$$

where $\mathbf{b}_i$ are the reciprocal lattice vectors of the moiré supercell (the summation runs over the 3 $\mathbf{b}_i$ vectors related by the $C_3$ symmetry). The product $\mathbf{b}_i \cdot \mathbf{r}$ equals $2\pi$ when $\mathbf{r}$ takes the value of the lattice vectors of the original moiré unit cell, and $n$ is an integer that commensurates the supermoiré length $L_{SM}$ with the original moiré length $L_M$ as $L_{SM} = nL_M$. Therefore, the function in eq. (7) allows to generate a modulated potential as the one shown in Fig. 6b for $L_{SM} = 5L_M$ in the original staggered honeycomb lattice. We can see that the cosine functions

of the supermoiré potential give rise to a 6 fold symmetry.

We now analyze the effect of a substrate on the twisted topological multiferroic. As noted above, a substrate induces a supermoiré potential in the twisted system as shown in Fig. 6b. When projected in the Wannier moiré orbitals, this modulation gives rise to an electrostatic potential that modulates the staggered Wannier honeycomb lattice that describes the twisted topological multiferroic. As shown in Fig. 6c, we introduce a modulated external field (eq. (7)) commensurate with a $30 \times 30$ ultracell, i.e. $L_{SM} = 30 L_M$, and analyze how the substrate might also lead to the emergence of topological excitations using the effective multiferroic Wannier Hamiltonian (eq. (6)). In our calculations the amplitude of the supermoiré potential $E_{SM}$ is normalized to the range $E_{SM}/t = [0, 1.3]$. This modulated potential creates regions with different topological invariant, $\mathcal{C} = 0$ on the maximum values of the potential and $\mathcal{C} = -1$ for the minimum values.[7] As a consequence, in the interface between these two topological regions zero energy states appear. In Fig. 6d, the local density of states at zero energy is plotted. We can observe that circular topological states emerge in the topological multiferroic. Since the origin of the interface modes is topological, a smooth reconstructions at the interface will not impact the boundary modes. This can be easily confirmed in Fig. 6d where the supermoiré potential induces regions with different topological invariant. In this case the boundaries have a different edge termination in each direction, but we can see that the circular boundary modes emerge without being affected by those different terminations. These zero energy states are commensurate with the modulation created by the substrate. Therefore, the substrate can also be seen as a source of topological excitations for the twisted topological multiferroic that we have studied.

## 6 Conclusions

To summarize, we have shown how a topological multiferroic order can emerge in twisted transition metal dichalcogenide bilayers. The staggered honeycomb lattice produced by the moiré system can be described by an interacting Wannier Hamiltonian with on-site and first neighbor Coulomb interactions. We have shown that, at quarter-filling, on-site interactions lead to a spin polarized system displaying a magnetic order, while first neighbor interactions promote a charge order leading to a spontaneous electric polarization. As a result, the combination of competing repulsive interactions leads to a multiferroic order displaying simultaneously ferroelectric and ferrimagnetic orders. A strong magnetoelectric coupling emerges due to the coupling between charge and spin degrees of freedom promoted by the competing interactions. We further showed that the inclusion of spin-orbit interactions associated to mirror symmetry breaking leads to a topologically non-trivial multiferroic order. We showed that the topological multiferroic displays topological excitations at the different ferroic domain walls, both in the spin and charge sectors. In particular, we have shown that external magnetoelectric control of these topological excitations can be achieved with external electric fields. Finally, by including the impact of an underlying substrate in the moiré system, we showed the emergence of topological excitations created by the supermoiré on the twisted topological multiferroic. Our findings put forward twisted dichalcogenides as a promising platform to engineer a topological multiferroic order. Finally, our results pave the way to achieve magnetoelectrically-tunable topological excitations, providing a starting point towards the potential use of topological mul-

---

[7]The Chern numbers in the different regions are taken as the ones that would correspond in the uniform limit for the corresponding local values of the parameters. Therefore, they are not explicitly computed locally for the modulated system. This would be formally possibly using a Green's function formalism. Nonetheless, for big enough domains, as those considered in our manuscript, the local Chern number in the moire can be directly inferred from its value in the uniform case with the associated local Hamiltonian parameters.

tiferroic modes in quantum technologies.

## Acknowledgements

We acknowledge the computational resources provided by the Aalto Science-IT project, and the financial support from the Academy of Finland Projects No. 331342, No. 336243 and No 349696, and the Jane and Aatos Erkko Foundation. We thank P. Liljeroth and M. Amini for useful discussions.

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
