# Peer review of "Topological multiferroic order in twisted transition metal dichalcogenide bilayers"

_SciPost Physics, doi:SciPost Phys. 13, 052 (2022)_

## Round 1 · Referee Report · Anonymous (Referee 1) · 2022-6-1

Report

The authors study topological multiferroic order in twisted transition dichalcogenides bilayers (TMDs) in the presence of spin-orbit interactions and applied electrical field. Starting from a model Hamiltonian that captures nearest-neighbor hopping between the AB/BA sites forming an effective honeycomb lattice, the authors analyze spin-/charge excitations (SDW/CDW) that emerge due to on-site and nearest-neighbor interactions within a self-consistent mean-field theory. They demonstrate that spin-orbit interactions render this model a topological multiferroic with non-vanishing Chern Number whose magneto-electrical coupling can further be tuned by the application of electrical fields. The topological character manifests in the existence of interface modes between either ferrimagnetic (FM), ferroelectric (FE) or multiferroic (FEM) domains. These findings are argued to exist in super-moire potentials arising from twisted TMDs encapsulated in substrates, e.g. hBN, with slightly different lattice constant. The authors demonstrate that the additional moiré potential leads to the formation of different topological regions of the multiferroic and hence naturally to the existence of topological interface modes.

The realization of topological states in twisted TMDs with strong intrinsic spin-orbit coupling represents an uprising field of research beyond the physics of twisted multilayer graphene and aligns with recent theoretical and experimental efforts. The discussion of topological multiferroic order is novel and adds to the toolbox of correlated topological phases that can be realized via moiré engineering. The connection to the super-moiré potential as possible candidate to host topological interface modes displays an interesting avenue, in particular because such effects have been observed experimentally in twisted trilayer graphene (TTG) to have major impact on correlated quantum states. The manuscript is soundly written and easy to follow for the reader. I therefore recommend publication as long as the following (minor) points are addressed properly:

Some technical aspects could be described in more detail to provide further clarification or to eventually allow for easier reproduction of the results, especially since there is no Appendix/Supplementary Material. Some aspects I came across are the following:

1.) Concerning the TMD model Hamiltonian: The nearest-neighbor honeycomb model has been widely studied in the literature within, e.g. mean-field approximation, functional renormalization group studies, quantum Monte Carlo etc. Most of these studies focus to half-filling/filling in the vicinity to the van-Hove singularity. Could the authors clarify how quarter-filling in their manuscript is related to the position of the vHS? This would make it easier to compare with existing results.

2.) The authors write: „The interacting model is solved using a self-consistent mean-field procedure including all the Wick contractions, including magnetic symmetry breaking, hopping renormalization, and charge order.“ I would prefer to avoid the statement „all Wick contractions“ as e.g. superconducting order is not accounted for as another possible Wick contraction or do the authors also account for this kind of instability, even though it may be irrelevant for purely repulsive interactions? Furthermore, what are the technical details of the mean-field analysis? How many moiré unit cells were taken into account (or what momentum resolution was used to sample the Brillouin zone) for the self-consistent procedure? Were the calculations performed in real- or momentum-space? Did the initial guess for the self-consistent procedure already include multiferroic order or was the initial state chosen randomly?

3.) Concerning the results of the mean-field analysis: It is interesting that for U/t<4 and up to V/t<2 no charge-density wave is present, but the latter only emerges for U/t>4 in the multiferroic phase. Is there an intuitive picture to understand why the formation of CDW order is diminished at U/t=0?

4.) The authors write: „Since the origin of the interface modes is topological, small reconstructions at the interface will not impact the boundary modes“. For simple graphene, the edge termination (zig-zag/armchair) is crucial for the existence of edge states. Have the authors tried different edge terminations for the L/R domains and found invariant results?

5.) How is the momentum dependence of the interface spectral function A(w, k_x) related to the sketch in Fig. 5 a) and what is the length of the domains in numerical simulations? From the sketch, I would intuitively expect that the system is periodic in y-direction and the interface separates the junction into the left/right domain along x. Then the momentum k_y would describe the bulk dependence. Maybe it would be helpful to add a small coordinate system in Fig. 5a) or to add a more detailed instruction why/how the spectral function A(w, k_x) is computed.

6.) Concerning the super-moiré potential in the last section: Can the authors give the functional form of the moiré potential and how it couples to the Wannier moiré orbitals? The potential looks to preserve the D_6h symmetry of the original honeycomb lattice, though the LDOS in the multiferroic state only seems to preserve D_3h. Is that due to the inversion-breaking Rashba SOC term? Furthermore, how are the Chern Numbers calculated for different regions of the super-moiré? Was the set of parameters for the mean-field Hamiltonian (m, Delta_z, lambda_R) the same as in the section before?

I hope the authors appreciate the comments given above, which should not distract from the high quality of the manuscript.

Typos:

1.) Introduction: „which may drive a non-trivial topological character in a the moire system[16].“
2.) Manuscript: „moire“. In the literature (and for all cited References) „moiré“ seems to be the usual term, but I leave this point up to the authors.
3.) Page 5: Up to this point, we have been based all the analyses on a full selfconsistent solution of the interacting Hamiltonian of eq. (1).
4.) Page 6: and acts as an independent topological source.
  • validity: high
  • significance: top
  • originality: top
  • clarity: high
  • formatting: excellent
  • grammar: good

Author:  Adolfo Otero Fumega  on 2022-07-09  [id 2645]

(in reply to Report 1 on 2022-06-01)

Please, find attached the response to this report.

Attachment:

Response_1.pdf

---

## Round 1 · Referee Report · Anonymous (Referee 2) · 2022-6-2

Strengths

See attached report

Weaknesses

See attached report

Report

See attached report

Requested changes

See attached report

  • validity: ok
  • significance: ok
  • originality: ok
  • clarity: ok
  • formatting: good
  • grammar: excellent

Author:  Adolfo Otero Fumega  on 2022-07-09  [id 2646]

(in reply to Report 2 on 2022-06-02)

Please find attached the response to this report.

Attachment:

Response_2.pdf

---

## Round 1 · Referee Report · Anonymous (Referee 3) · 2022-6-8

Strengths

1- A major and important topic of condensed matter in 2D materials.

2- The numerical method used is adapted and seems to me correctly applied.

3- A thorough, detailed and well explained study.

Weaknesses

1- Is the model used (Wannier Hamiltonian) relevant to study magnetism of twisted transition metal dichalcogenide (TMD) bilayers ? In particular, this approach does not directly take into account the specific roles of the d-orbitals (Transition Metal) and p-orbitals (Chalcogenides) which do not have the same characteristics in terms of magnetization.

2- This study only considers magnetism in the z direction, but the magnetic moment may be in favor of the xy plane rather than the z direction, especially for a system that is not half filled.

Report

This paper presents a detailed numerical study of the magnetism of low angle twisted bilayer TMD, using a Wannier Hamiltonian. It is shown how a topological multiferroic order can emerge in this system at quarter-filling. The competitive effects of on-site and first-neighbor interaction terms are studied in detail. As a result, a multiferroic order displaying simultaneously ferroelectric and ferrimagnetic orders can be obtained. Spin-orbit interactions induce a topologically non-trivial multiferroic order, and topological excitations can be control by and external electric fields. The author have also studied the impact of an underlying substrate in the moiré system by considering super-moiré cell, and thus show the emergence of topological excitations. This theoretical work is very rich and brings advances in a complex and new subject the physics of the twisted bilayer semiconductor. It is well presented and I think it deserves to be published in SciPost Physics.

Requested changes

To reinforce the relevance of this work it seems important to me to discuss the following points in more detail.

1- Wannier Hamiltonian model is widely used in the literature to study the electronic properties of the low-energy states of twisted bilayer. However, its relevance to the study of magnetism is not obvious. The low energy eigenstates, that are well simulated by the Wannier Hamiltonian (without interaction), are complex and localized on a large number of atomic orbitals (mainly d-obitals of metal). When the interaction terms are on, the local magnetic order (at the atomic scale) can be complex. Is it well simulated by Wannier states? It would be interesting for the authors to discuss the consequences of their results for the magnetization of the d-orbitals of metal.

2- If I am not mistaken, the xy plane magnetization is not include in the calculations. The authors should justify that point, especially for a calculation that is not at half-filling.

3- The authors focused on the quarter-filling case. It would be nice to justify this choice a little more. Is it also possible to discuss, at least qualitatively, the cases of other fillings?

4- The SciPost latex style should be used to produce the manuscript.

  • validity: high
  • significance: good
  • originality: good
  • clarity: high
  • formatting: good
  • grammar: -

Author:  Adolfo Otero Fumega  on 2022-07-09  [id 2647]

(in reply to Report 3 on 2022-06-08)

Please find attached the response to this report.

Attachment:

Response_3.pdf

---

## Round 2 · Referee Report · Anonymous · 2022-7-19

Strengths
1- The manuscript describes an interesting model, currently being intensively studied.
2- The manuscript presents an extensive study of the phase diagram of the model,
3- The manuscript discusses interesting phases, with novel and non trivial properties.
Weaknesses
1- The manuscript discusses generic properties of the model, with little attention to the actual size of the effects which may be, or may be not, observed.
2- The manuscript is vague when discussing the range and relative strength of the parameters of the model.
3- The results are restricted to a narrow range of bands (1) and filling factors (1/4, perhaps also 3/4).
Report
The new version of the manuscript includes a clearer and more in depth discussion of the physical issues addressed in it. I understand that further improvements would imply a substantial change in scope. The manuscript, as it is, is interesting, and it can be published.

---

## Round 2 · Author Response

We are resubmitting our manuscript "Topological multiferroic order in twisted transition metal dichalcogenide bilayers", for consideration in SciPost Physics.
Our manuscript was reviewed by Referee 1, Referee 2 and Referee 3. Referee 1 recommended our manuscript for publication after addressing minor points and highlighting the high quality of the manuscript. Referee 2 stated that the results are interesting and deserve publication, providing useful comments to improve the clarity of the article. Referee 3 reported that our theoretical work is very rich and brings advances in a complex and new subject, it is well presented and deserves to be published in SciPost Physics.
In our revised version, we have addressed all the comments of the Referees and modified our manuscript accordingly.
In this work, we provide a theoretical route to achieve a multiferroic order in a twisted system. Furthermore, we show how to introduce a non-trivial topological character in the system, and how to exploit it along with the multiferroic behavior to achieve a magnetoelectric control of topological excitations. Given all the points above, we hope that you consider our manuscript suitable for publication in SciPost Physics.
Yours sincerely,
Mikael Haavisto, Jose L. Lado and Adolfo O. Fumega

---

## Round 2 · List of Changes

- 6 new footnotes (2-7) were added addressing many different points raised by the referees.
- In section 2. Model after eq. (1), the discussion of the model for the twisted system was improved including the response to the comments made by the referees to this point.
- At the beginning of section 3, the first paragraph was added to motivate the choice of the quarter-filling case and to compare it with other fillings. In the third paragraph, a discussion about the interplay between the coulomb parameters to get a multiferroic order was added. The fourth paragraph was added providing estimates of the values of the electric dipole and magnetization that one would obtain in this twisted multiferroic.
- In section 4, the last paragraph was added providing a discussion on the value of the Rashba spin-orbit coupling required to achieve a topological multiferroic.
- In section 5, Fig. 5 was modified accordingly to the referee's comment. In subsection 5.1 at the end of the second paragraph, a discussion on the interface smoothness between different domains and the robustness of the topological states was included. In subsection 5.2, a functional form for the supermoiré potential was included (eq. (7)) with the corresponding explanation. In the final paragraph, a discussion on the interface smoothness between different topological regions was added.
- Overall, we have improved the readability of the article, corrected the typos pointed out by the referees and change the artice style to SciPost style.

---

## Editorial Decision

published